# Traceable Characterization of Nanomaterials by X-ray Spectrometry Using Calibrated Instrumentation

**DOI:** 10.3390/nano12132255

**Published:** 2022-06-30

**Authors:** Burkhard Beckhoff

**Affiliations:** Physikalisch-Technische Bundesanstalt, Abbestraße 2-12, 10587 Berlin, Germany; burkhard.beckhoff@ptb.de

**Keywords:** traceability, characterization, elemental analysis, speciation, nanostructures, nanoparticles, XRF, GIXRF, XAFS, XES

## Abstract

Traceable characterization methods allow for the accurate correlation of the functionality or toxicity of nanomaterials with their underlaying chemical, structural or physical material properties. These correlations are required for the directed development of nanomaterials to reach target functionalities such as conversion efficiencies or selective sensitivities. The reliable characterization of nanomaterials requires techniques that often need to be adapted to the nano-scaled dimensions of the samples with respect to both the spatial dimensions of the probe and the instrumental or experimental discrimination capability. The traceability of analytical methods revealing information on chemical material properties relies on reference materials or qualified calibration samples, the spatial elemental distributions of which must be very similar to the nanomaterial of interest. At the nanoscale, however, only few well-known reference materials exist. An alternate route to establish the required traceability lays in the physical calibration of the analytical instrument’s response behavior and efficiency in conjunction with a good knowledge of the various interaction probabilities. For the elemental analysis, speciation, and coordination of nanomaterials, such a physical traceability can be achieved with X-ray spectrometry. This requires the radiometric calibration of energy- and wavelength-dispersive X-ray spectrometers, as well as the reliable determination of atomic X-ray fundamental parameters using such instrumentation. In different operational configurations, the information depths, discrimination capability, and sensitivity of X-ray spectrometry can be considerably modified while preserving its traceability, allowing for the characterization of surface contamination as well as interfacial thin layer and nanoparticle chemical compositions. Furthermore, time-resolved and hybrid approaches provide access to analytical information under operando conditions or reveal dimensional information, such as elemental or species depth profiles of nanomaterials. The aim of this review is to demonstrate the absolute quantification capabilities of SI-traceable X-ray spectrometry based upon calibrated instrumentation and knowledge about X-ray interaction probabilities.

## 1. Introduction

The functionality or toxicity of a nanomaterial depends on the spatial distribution of its elemental and species concentrations or mass depositions. While analytical chemistry establishes traceability chains based on pure substances or certified matrix reference materials, physical metrology links each measurement’s result, including its associated uncertainty, back to the International System of Units (SI) [1]. At the nanoscale, analytical characterization techniques must provide reliable information on elemental, species, or coordination mass deposition, i.e., the number of atoms in a particular chemical binding or coordination state per unit area, in order to allow accurate correlations with the nanomaterial’s functionality or toxicity. The directed development of nanomaterial functionalities as well as the assessment of nanoobject toxicities require this kind of information.

Steadily increasing needs for the provision of more complex functionalities of nanomaterials and the adaptable usage of various technological processes in one device or in a set of closely adjacent devices call for a transition of empirically based knowledge to a more complete understanding of the processes generating specific device functionalities, such as conversion efficiencies or tunable interfacial properties [2]. To probe these underlying chemical and physical properties reliably, traceable characterization methods that provide sufficient sensitivity for the chemical and physical quantities of interest in nanomaterials are required. In general, this requires adaptions of characterization techniques to the nano-scaled dimensions of the samples regarding the spatial dimensions of the probe and the instrumental discrimination capability. In view of the drastic lack of reference materials at the nanoscale [3], the quantification reliability of analytical methods based on reference materials, e.g., for the compensation of missing knowledge on instrumental or experimental parameters of the techniques, is considerably affected.

The challenge of missing reference materials at the nanoscale can be addressed by an alternate route for traceability. The German National Metrology Institute (NMI), Physikalisch-Technische Bundesanstalt (PTB), established X-ray spectrometry (XRS) as an X-ray analytical method traceable to the SI. For this purpose, all instrumental and experimental parameters need to be well determined, allowing their respective contributions to the uncertainty of the analytical result in the form of an elemental mass deposition to be known [4]. In addition, good knowledge on the values and related uncertainties of the X-ray interaction probabilities, i.e., the atomic X-ray fundamental parameters (FPs) [5], has to be ensured either by means of appropriate data in the literature or dedicated experimental or theoretical determinations. In the same way as traceable XRS, such experimental FP determinations require instrumentation that has been calibrated with respect to all relevant parameters, such as the incident radiant power, spectral purity and angle of incidence of the excitation radiation, the angle of observation and the solid angle of detection, as well as the spectral response behavior and efficiency of the wavelength or energy-dispersive detection system employed to record the element-specific X-ray fluorescence radiation. Most instrumental parameters can be determined based upon principles of X-ray radiometry [6]. On the other hand, the careful assessment of existing FP literature data as well as well-coordinated activities to reveal FP data with reduced uncertainties using modern methodologies have been organized since 2008 by an international consortium of academic and industrial key players, and have been disseminated by a series of annual workshops as well as X-ray fundamental parameter roadmap documents [7]. Coordinated FP-determination activities involve improved experimental and theoretical methods based upon novel kinds of instrumentation, ultra-thin one- or two-elemental specimens, and advanced algorithms [5,8,9]. Improved FP values allow us to reduce the uncertainties of analytical XRS measurement results for both the chemical and physical traceability chains to the SI.

The nano-scaled dimensions of nanomaterials make it more difficult to ascertain many key analytical parameters of characterization techniques. One main problem is that the amount of substance to be probed can be several orders of magnitude lower than is the case for pure substances or conventional reference matrix materials. This requires a high dynamic of both analytical detection sensitivity and quantification reliability. Additionally, significant signal crosstalk can be induced by the substrate response to the probe, thus posing challenges to both the analytical discrimination capabilities and to the spatial dimensions of the probe in terms of beam profile or beam penetration depths. The variation of XRS operational parameters enables one to meet those requirements associated with the characterization of nanomaterials. The usage of tunable monochromatic and linearly polarized excitation radiation drastically reduces spectral background in the regions of interest, thus enhancing the XRS detection sensitivity. Beam-focusing optics or grazing incidence conditions with very low penetration depths can reduce spectral substrate contributions, thus improving the discrimination capability of XRS. The same holds when employing higher-resolution crystal or grating spectrometers instead of energy-dispersive X-ray detectors.

The use of well-known and tunable synchrotron radiation (SR) [4,6] as monochromatic and polarized excitation radiation ensures highly sensitive SI-traceable XRS applications. For incident angles below the angle of external total reflection, the penetration depth of the incident beam is limited to a few nanometers, rendering this operational configuration ideal for the determination of elemental or species surface contamination [10] on very flat substrates with roughness values in the sub-nanometer range. When tuning the incident angle above the angle of external total reflection, the penetration depth considerably increases and quantitative analytical access to near-surface interfaces [11,12], diffusion or dopant depth profiles [13], and nanolayered systems [12,14] is enabled. By tuning the photon energy of the excitation radiation across the absorption edge of an element of interest, i.e., by performing X-ray absorption fine structure (XAFS) spectrometry, complementary information on the chemical binding state [12] can be revealed. To reveal depth-resolved chemical information on interfaces or nanolayered systems [11,12], XRS experiments at suitable sets of incident angles must be performed to enable a differential quantification approach. For very flat substrates and nanolayers, one takes advantage of the nano-scaled structure of X-ray standing wave (XSW) field distribution, which strongly depends on the angle of incidence as well as on the spatial elemental distribution of the nanomaterial of interest. The XSW field modifies the effective excitation intensity in all XRS experiments at very flat surfaces or interfaces. Such grazing-incidence XRS approaches allow for the quantitative characterization of different surface functionalization [15,16] and of biomedical materials [17,18]. Based upon specific sample environments, XRS could be applied to the chemical analysis of liquid samples [19,20]. Recent round robin activities in XRS under total reflection conditions have confirmed its quantification reliability for both chemical as well as physical traceability chains [21]. XRS under grazing incidence or shallow-angle observation can reveal both analytical and dimensional information from layered systems, nanostructures [22], and particles deposited on flat substrate surfaces. In the following sections, the basic principles, different measurement configurations, and selected applications of SI-traceable XRS will be described.

## 2. Reference-Free X-ray Fluorescence Analysis

The elemental analysis of physically traceable XRS is known as reference-free X-ray fluorescence (XRF). The term reference-free has been defined as a quantification approach not relying on any reference material or calibration samples. At PTB, reference-free XRF has been enabled by using tunable monochromatized synchrotron radiation with both well-known spectral purity and radiant power for specimen excitation. To ensure well-known experimental XRF configurations, several UHV instrumentations have been built and characterized by PTB. These instruments allow for the accurate definition of the angles of incidence and observation, as well as the accurate determination of the effective solid angle of detection. Photodiodes and energy-dispersive silicon drift detectors (SDD) are calibrated by different radiometric techniques [6] in the PTB laboratory at the SR facility BESSY II. When measuring the radiant power of tunable SR by means of a calibrated photodiode, the absolute efficiency and response functions of an SDD can be determined at different photon energies of interest. For reference-free XRF and SI-traceable XRS, two well-characterized beamlines for monochromatized SR are employed: a plane grating monochromator (PGM) beamline for undulator radiation and a four-crystal monochromator (FCM) for bending magnet radiation.

Figure 1 illustrates basic geometrical arrangements in reference-free or SI-traceable XRF at different angles of incidence: total reflection X-ray fluorescence (TXRF) at shallow angles below the critical angle of total reflection θ_crit_, grazing incidence X-ray fluorescence (GIXRF) at angles up to about four times the value of θ_crit_, and conventional XRF in the 30° to 90° angular range. In PTB’s XRS instrumentation, different photodiodes are used to record the radiant power of the incident, transmitted, or reflected radiation. The transmittance through a thin sample can contribute to selected FP determinations, while the radiation being reflected at a very flat sample into a slit diode allows for alignment purposes and for revealing dimensional information on roughness values or thin layer thicknesses and densities.

The XRF measurand is the detected fluorescence intensity of discriminable elemental fluorescence lines that can be converted into the related quantity of the mass *m* of the element *i* per unit area *A* by means of the Sherman equation. Other expressions of this quantity are the aerial mass, the mass thickness, or the mass of atomic surface density. The quantification of elemental mass depositions *m_i_/A* by reference-free XRF requires the knowledge of all relevant instrumental, experimental, and atomic fundamental parameters. The concentration *C_i_* of the element *i* corresponds to the elemental mass deposition *m_i_/A* divided by the sum of all elemental mass depositions in the sample. In a homogeneous multi-elemental sample of thickness *d*, the value of *C_i_* can be calculated according to the following Sherman-type equation:$$C_{i}=\frac{P_{i}}{P_{0}\tau_{i,E_{0}}Q\frac{\Omega_{det}}{4\pi }\frac{1}{\sin\Theta }\frac{1-\exp(-\mu_{tot,i}d)}{\mu_{tot,i}}}$$
where:$E_{0}$ is the incident radiation photon energy;$P_{0}=S_{0}/s_{diode,E_{0}}$/$E_{0}$ is the incident photon flux;$S_{0}$ is the signal of the photodiode recording the incident radiant power;$s_{diode,E_{0}}$ is the spectral responsitivity of the photodiode;*θ* is the incident angle;*E_i_* is the photon energy of the fluorescence line *l* of the element *i*;*R_i_* is the detected count rate of the fluorescence line *l* of the element *i*;$\varepsilon_{det,E_{i}}$ is absolute SDD efficiency at the photon energy *E_i_*;$P_{i}=R_{i}/\varepsilon_{det,Ei}$ is the intensity of the fluorescence line *l* of the element *i*;$\tau_{i,E_{0}}$ is the photo electric cross section of the element *i* at the photon energy $E_{0}$;$\mu_{S,E}$ is the absorption cross section of the sample *s* at the photon energy *E*.
$$\mu_{tot,i}=\mu_{S,E_{0}}/\sin\theta +\mu_{S,E_{i}}/\sin\psi_{}$$
where:$\psi_{}$ is the angle of observation with respect to the sample surface;*Ω_det_* is the effective solid angle of detection defined by both a calibrated SDD aperture placed at a well-known distance from the sample and the incident beam foot print on the sample surface;*ω_Xi_* is the fluorescence yield with respect to the (sub)shell or orbital *Xi* (of the element *i*);*g_l,Xi_* is the transition probability of the fluorescence line *l* associated with *Xi*;*j_Xi_* is the ratio of the photo electric cross section of *Xi* to the sum of the photo electric cross sections of all other shells that can be ionized at $E_{0}$;*Q* = *ω_Xi_ g_l,Xi_* (*j_Xi_* − 1)/*j_Xi_*.

In the case of very flat samples excited under grazing incidence conditions, the modulation of the incident radiant power by the XSW intensity [11] has to be included as an additional factor. Secondary and tertiary excitation channels within a homogeneous layer or bulk sample, as well as intra-layer excitation and absorption effects can be likewise included as indicated in detail in the literature [23,24]. Contributions of secondary excitation fluorescence [24] can exceed 20%, depending on the elements involved along with the spatial composition and dimensions of the layered system, the excitation energy, and the angle of incidence. For inhomogeneous matrices, analytical expressions of elemental concentrations can only be given when descriptions of the spatial distributions of the main matrix constituents, e.g., in the form of a parametrized elemental depth profile, are available. Otherwise, Monte-Carlo-based XRF modelling may contribute to overcoming a lack of sufficient a priori knowledge concerning the spatial distribution of main matrix elements or, when including multiple scattering effects for which analytical expressions are more difficult to deduce, in spectral decomposition.

Figure 2 describes the different configurations and fields of applications of reference-free XRS for well-known experimental parameters with sufficient a priori knowledge of the spatial distribution of the main matrix constituents, as is the case for homogeneous and layered samples. Many new materials, e.g., in the semiconductor industry, are essentially thin layered samples at the nanoscale. The tunability of SR across the absorption edge of a main matrix element allows for chemical speciation by XAFS. For nanolayer or interfacial speciation purposes, both the incident photon energy and incident angle need to be varied in a pre-selected manner [11,12] in order to take advantage of the related tunable intensity of the nano-scaled XSW field. Thereby, one can either perform species depth profiling or interfacial speciation in nanolayered systems.

With respect to characterizations of nanoparticles and environmental aerosols, both TXRF [25] and GIXRF [26] have shown their analytical capabilities to quantify elemental mass depositions of size-fractionated nano- and microparticles collected by means of cascade impactors on flat substrates. A reference-free GIXRF quantification mode [26] allows one to determine the angular ranges of validity for the correct quantification of specific elemental mass depositions. Figure 3 shows the angular GIXRF responses for different nitrogen mass depositions in conjunction with the corresponding reference-free GIXRF quantification, clearly allowing for the identification of the angular regimes providing constant mass depositions. Engineered nanoparticles, often having small size distributions, are ideal nanoobjects to be characterized by GIXRF [27] with respect to surface coverage and particle composition.

## 3. High-Resolution X-ray Spectrometry

Besides revealing information about elemental mass depositions and related chemical binding states of nanomaterials by means of XRF and XAFS, complementary chemical information can be revealed using high-resolution X-ray emission spectrometry (XES), which often provides a different discrimination capability depending on the specific valence states of compounds. XES can substantially contribute to the further development of complex nanomaterials with distinct chemical properties. In the field of catalysis, for example, the identification and quantification of active sites is important for a more thorough understanding of catalytic systems. Using a physically traceable XRS approach based on calibrated instrumentation and the knowledge of atomic fundamental parameters, absolute elemental and species mass depositions can be derived. For this purpose, PTB built a compact and calibratable wavelength-dispersive spectrometer [28,29] for XES in the photon energy range of 2.4 keV to 18.0 keV. This von Hamos spectrometer has a total length of 1450 mm when being mounted at a UHV XRF instrumentation, and allows for either an individual or combined usage of one or two full cylindrical highly annealed pyrolytic graphite (HAPG) crystals having a radius of only 50 mm and a deposited HAPG thickness of 40 µm. Employing these HAPG cylinders as dispersive elements in a von Hamos geometry, large solid angles of acceptance resulting in high detection efficiency are realized while preserving a moderate-to-high resolving power.

Over the last decade, spectrometers in von Hamos geometries have become quite popular in both synchrotron radiation facilities as well as in laboratories outside large-scale facilities [30,31,32,33,34,35,36]. Some of these spectrometers use HAPG, which is a synthetic carbon-based mosaic crystal with very low values of angular distributions of these mosaic blocks (mosaic spreads below 0.1°). Thus, the Bragg reflection of X-rays at HAPG lays in between dynamic and kinematic diffraction regimes, which still ensures a high integrated reflectivity. HAPG can be deposited to substrates with small radii of curvature down to 50 mm [37,38].

The characterization of the PTB von Hamos spectrometer included the determination of its energy scale by means of known fluorescence line energies, or by transferring the FCM energy scale in elastic scattering experiments. With respect to the von Hamos spectrometer response behavior, different influence parameters such as the effective X-ray source size (i.e., the excited sample volume) and its different operational modes, including Bragg reflections at only one HAPG cylinder, subsequent Bragg reflections at two HAPG cylinders, and second-order Bragg reflection at one HAPG cylinder, have been evaluated. Furthermore, its absolute detection efficiency could be derived in a reference-free XRF experiment by means of a comparison to a calibrated SDD having a well-known solid angle of detection. The chemical speciation and discrimination capabilities of the von Hamos spectrometer have been investigated using various transition metal compounds.

The calibration of the instrumental response of the wavelength-dispersive spectrometer enables an accurate determination of binding state-related structures in transition metal compound spectra [39], as shown exemplarily in Figure 4, thus enabling reliable identification and quantification capabilities. The HAPG cylinder length of 20 cm offers the advantage of recording spectra of several hundreds of eV width, allowing one to determine the energy positions and transition probabilities associated with one absorption edge in only one measurement [40], thus reducing the risk of combining separate energy scales.

There are currently only a few dedicated scanning X-ray microscopes at SR facilities that offer XRF analysis in combination with scanning transmission X-ray microscopy (STXM) and lateral resolutions in the nanometer regime [41,42,43,44,45,46,47,48,49,50,51,52]. In general, XRF quantification is based on either certified reference materials or calibration samples, i.e., on chemical traceability principles. In order to achieve high spatial resolutions for physically traceable XRF in the nanometer regime, PTB built an add-on arrangement for the sample scanning and alignment stage of one of its UHV XRS instruments. This piezo-stage-based add-on set-up [53] allows one to place an Au zone plate with an integrated beam stop, an order sorting aperture, and nano-scaled samples on a single plate, which is mounted as one piece on the UHV XRS instrument manipulator in order to reduce vibrational pick-up. Initial transmission and XRF experiments were performed at the PGM beamline at a photon energy of 1500 eV, revealing spatial resolutions of 90 nm and 140 nm, respectively. Follow-up nm XRF experiments aimed to establish a reliable quantification scheme based upon both the physically traceable approach using calibrated instrumentation, as well as the joint evaluation of sets of nm XRF experiments on adjacent lateral positions of structured nanomaterials.

## 4. Determination of Atomic Fundamental Parameter

The quantitative analysis of XRF and particle-induced X-ray emission spectroscopies requires good knowledge of the atomic FP values and optical constants of the elements of interest. The uncertainties of FP data tabulated in the literature are partially rather large, in particular for low-Z elements or L- and M-shell transitions, calling for new experimental determinations of accurate FP data. For this purpose, recent FP experiments have taken advantage of several instrumental improvements such as SR beamlines providing monochromatic radiation of high spectral resolution, SDDs with better energy resolution than the energy resolution of previous Si(Li) detectors, and the availability of free-standing one- or two-elemental foils with thicknesses down to 70 nm [4]. Those FP experiments aimed to determine low-Z fluorescence yields, subshell photoionization cross sections (PCS) [5] and Coster–Kronig factors [54]. The values of recently determined PCS confirmed that previously used jump ratio approaches, as recommended by the International Union of Pure and Applied Chemistry (IUPAC), are often not accurate. With respect to XRF quantification, these FP values have a direct impact on the revealed elemental mass depositions of nanomaterials. Figure 5 shows the energy dependence of experimentally determined Pd-L subshell PCS in comparison to the theoretical data. Figure 6 shows the spectral deconvolution of two related XRF spectra by means of detector response function. In order to reduce the uncertainties of the PCS, transmission measurements of the thin Pd samples were performed to experimentally determine the self-absorption correction factors also used for spectral deconvolution.

For reference-free XRF in the soft X-ray range, the uncertainties of FP associated with L-edges are crucial for the total uncertainty of the quantification. Using very thin samples, the fluorescence yields and Coster–Kronig transition probabilities for the L-edges of Ga were determined with considerably reduced uncertainties [55]. Due to the high absorption of radiation in the soft X-ray range, it is necessary to reduce the thickness of the samples for transmission measurements with sufficient dynamics. Transmission measurements are crucial for performing absorption correction without having to rely on literature database values of the mass attenuation coefficients.

The French NMI at the Laboratoire National Henri Becquerel (LNHB) uses tunable SR at SOLEIL for the determination of various mass attenuation coefficients. Along with various collaboration partners in different institutional and industrial metrology research projects, several elements of soft and hard X-ray ranges have been studied. Emphasis has been placed on ensuring reliable uncertainties of FP values. Thereby, complementary theoretical and experimental approaches have been considered [9,56,57,58]. Some of the results of these studies do not agree with FP data previously published by other groups within their respective uncertainties. These deviations have been evaluated by the international Fundamental Parameter Initiative www.EXSA.hu/fpi.php (accessed on 31 January 2022).

At PTB, a calibrated wavelength-dispersive grating spectrometer has been used to derive FP values such as transition probabilities [59] in the soft X-ray range and to contribute to a better understanding of the electronic structures of light elemental compounds. In a collaboration between the U.S. NMI NIST and PTB, joint high-resolution soft X-ray experiments such as XES, resonant inelastic X-ray scattering (RIXS), and XAFS of different nitrogen-containing compounds were performed at BESSY II, the results of which were compared to calculations using the OCEAN BSE code for core-level spectroscopy [60,61,62]. Different first-principle-calculation approaches for the simulation of X-ray spectral information provided by XES, RIXS and XAFS have been considered and can provide a theoretical model-based understanding of observed experimental phenomena. Currently, PTB has started the commissioning of a new wavelength-dispersive VLS grating spectrometer based on a slit-less Hettrick–Underwood geometry that considerably enhances both the detection efficiency and the energy resolution.

## 5. Hybrid Metrology—Determination of Dimensional and Analytical Information of Nanomaterials

About one decade ago, so-called hybrid metrology surfaced to ensure the more reliable characterization of nanomaterials. The term hybrid metrology means that two or more different characterization techniques are employed on an object of interest, such as a nanomaterial, and, if necessary, on its duplicates, to gain more information, to reduce uncertainties, or to reduce measurement times. In general, different measurement techniques are associated with different measurands, even when aiming for the same physical or chemical quantity, e.g., in the dimensional, electrical, or analytical fields. One of the first industrially relevant applications of hybrid metrology was in the nanotechnology sector where the critical dimension (CD) metrology of printed 3D nanostructures employed different techniques such as optical CD scatterometry, CD scanning electron microscopy, and CD atomic force microscopy to reveal improved dimensional information [63].

For different deviations of various measurement techniques for a particular physical or chemical quantity to be revealed by the measurements as an attribute of a phenomenon, body, or substance [64], a modelling-assisted conceptional and theoretical understanding of those deviations [65,66] can be expected to allow for the best use of hybrid metrology. To this respect, the NIST reference material No. 8011—‘gold nanoparticles (nominal 10 nm diameter)’ [67,68]—illustrates that the measurement results of different characterization techniques may not always overlap within their respective uncertainties.

With respect to the modelling-assisted understanding of the measurements of nanomaterials, some aspects of the characterization techniques become crucial when combining them in a hybrid approach: both the penetration and information depths as well as the probing volumes of different techniques may differ, and, in addition, may not match well with some of the sample dimensions or the spatial sample heterogeneity. With the advent of machine learning algorithms that can handle large measurement data sets as quickly as the actual measurement times last, part of the missing knowledge needed for a complete modelling-assisted understanding of the measurement processes in hybrid metrology may be compensated for [69]. However, substantial knowledge on the physical relationship between the techniques’ measurands and the particular physical or chemical quantities of interest can further reduce uncertainties and allow for mutual validations of quantification schemes.

In many fields of modern nanotechnology, the reliable determination of spatial elemental and species distribution is crucial for R&D, as well as quality assurance processes during the manufacturing of layered or 3D nanostructures. With decreasing device dimensions and increasing structure complexity, the challenges to key parameters of characterization methods increase as well. In addition, it is crucial to reveal all dimensional and analytical information on such advanced nanomaterials in order to achieve a complete understanding of the dependence of their functionality on chemical and physical quantities.

When characterizing one-elemental nanolayers with both the dimensional technique of X-ray reflectometry (XRR) and the analytical technique of XRF, complementary information about the layer thickness d and density ρ, as well as the elemental mass deposition *m_i_/A*, can be deduced [70]. Here, the physical relationship between the particular quantities can be directly stated as *d*ρ = *m_i_/A*. The corresponding XRR and XRF results of 11 different single- and double-layer systems of Ni and Cu agree within their respective uncertainties [70]. When performing hybrid experiments using XRR and XRF simultaneously under grazing incidence conditions, one can take substantial advantage of knowledge concerning the physical relationship *d*ρ = *m_i_/A* as a free-parameter-reducing constraint for the data evaluation of several independent complementary measurements. Furthermore, additional information on combined dimensional and analytical quantities, such as the elemental depth profile within a nano-scaled layered system [71], can be revealed when varying the angle of incidence.

When combining XRR with reference-free GIXRF, direct access to the mass depositions (ρd) of the materials of interest is provided. This allows for a significant reduction in the degrees of freedom within the combined GIXRF-XRR modelling, and thus improves the characterization reliability. The combined reference-free GIXRF-XRR approach has been applied for the depth-resolving analysis of thin nano-laminate stacks of Al_2_O_3_ and HfO_2_ layers with total thicknesses in the sub-10-nanometer regime [71]. For the GIXRF modelling of these nanostructures, novel approaches for the calculation of the spatial distributions of XSW field intensities [22,72] based on 3D Maxwell equation solvers are necessary. Figure 7 shows the spatial XSW intensity distribution at a regular nanostructure that can be effectively tuned by varying both the grazing and azimuthal angles of the incident radiation with respect to the nanostructure. This local tunability of the XSW field with respect to the spatial elemental distribution of the nanostructure to be probed ensures the high discrimination capability of this hybrid method. A recent work [73] demonstrates the corresponding reconstruction capabilities of GIXRF analysis used at periodic nanostructures to reveal both detailed dimensional and analytical information. For the calibration of laboratory GIXRF instruments, i.e., for a combined determination of unknown experimental and instrumental parameters, specific nanolayered structures could be qualified by means of SR-based reference-free GIXRF analysis [74].

With respect to nanomaterial development and quality management applications in the semiconductor industry, it is also valuable to take advantage of the optical equivalence of grazing incidence and grazing emission XRF (GI- and GEXRF) principles. The GEXRF approaches have been successfully explored for more than two decades at various large-scale facilities and in academic laboratories [75,76,77,78,79,80,81], in particular regarding the characterization of nanolayered systems. To complement angular GIXRF studies [22,72] and related reconstruction works on nanostructures [73], GEXRF also allows for the scanning-free detection of the angular distribution of fluorescence radiation emitted by a nanostructure using position-sensitive detectors such as CCDs or CMOS devices [82].

## 6. Operando Metrology—Time-Resolved Determination of the Analytical Information of Batteries

The increasing demand for more efficient secondary electrochemical storage devices requires well-characterized battery systems. Every battery suffers from degradation effects which lead to capacity fading and life cycle reduction. With electrochemical methods, the fading can easily be monitored, but to understand the underlying chemical and physical properties, which are responsible for the capacity reduction, further spatial and time resolving analytic techniques are needed [83]. Using well-known tunable synchrotron radiation and calibrated XRS instrumentation, traceable XRF and XAFS investigations of the degradation mechanisms of lithium sulfur (Li/S) batteries are enabled. These degradation processes are associated with the formation of soluble polysulfides causing capacity fading, and limit the cycle life. The operando XRF method allows for the absolute quantification of the mass deposition of sulfur in dissolved polysulfides without the need for any calibration samples. Suitable cell designs, including sufficiently X-ray transparent entrance windows, enable the probing of polysulfides at both battery electrode sides for three full charge–discharge cycles, which leads to the simultaneous investigation of conversion reactions as well as transport mechanisms and, therefore, the possibility to evaluate polysulfide shuttle phenomena (see Figure 8). In carefully evaluating all XFAS spectra, taken about every 8 min under operando conditions, the time-resolved changes in average polysulfide chain length enable a deeper understanding of the capacity-fading processes [84].

With respect to energy density, the lithium-ion battery (LIB) currently defines the state of the art, but substantial lifetime enhancements call for novel material developments allowing the optimization of nano-scaled interfacial properties. LIBs comprise a graphite negative electrode and a layered transition metal oxide positive electrode. Nickel manganese cobalt (NMC) materials are among the promising candidates for positive electrodes, but challenges remain in terms of improving lifetime. Degradation mechanisms include irreversible phase changes, nickel–lithium site exchange, and the oxidation of lattice oxide leading to metal dissolution, but a full understanding of their interplay is lacking. Regarding aging processes, quantitative elemental and species analyses are important for identifying critical processes and for assessing the relevance of a specific process for the total capacity decrease. A main challenge in the investigation of aged battery materials is the lack of appropriate reference materials on which many analytical techniques rely with respect to their traceability. With the use of reference-free XRF, the total mass deposition of manganese deposited at the anode for an aged cell has been quantitatively determined under ex situ conditions. For 50 full cycles with elevated cut-off voltage, the capacity decreased by 12.5%, while up to 0.16% of cathodic manganese was found to be deposited in the anode [85].

## 7. Conclusions

The correlation of nanomaterials’ functionalities with their underlying physical and chemical properties is crucial for the further advancement and assessment of materials in development, manufacturing, and quality control processes. Physically traceable X-ray spectrometry methods can contribute to these nanoanalytical challenges by their high elemental and species sensitivity, combined with substantial discrimination capability tunable by a broad set of operational parameters. X-ray spectrometry allows—in combination with other analytical, dimensional, or electrical techniques—both hybrid and operando configurations. Such multimodal and time-resolved approaches can provide simultaneous access to various physical and chemical quantities, which is crucial for a more complete understanding of nanomaterials. Here, the spatial elemental or species distribution of a specific nanomaterial determine the fruitful application range of the various characterization techniques to be combined. Upcoming research challenges, such as the time-resolved understanding of degradation processes in batteries at interfacial locations, will certainly call for the combination of hybrid and operando metrology, including SI-traceable X-ray spectrometric techniques.

## Figures and Tables

**Figure 1 nanomaterials-12-02255-f001:**
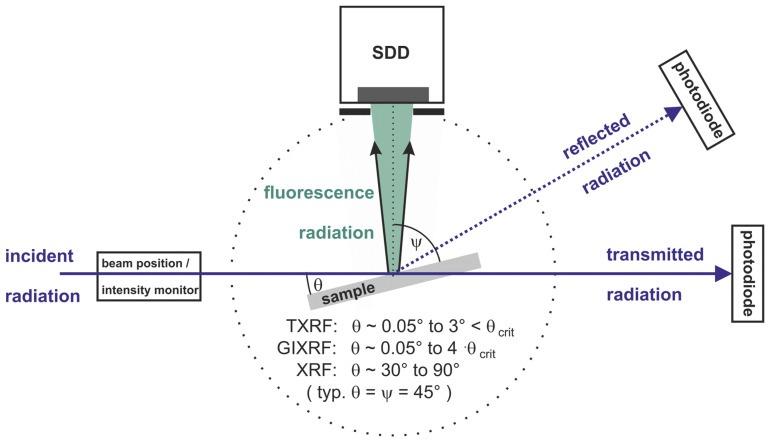
Different experimental configurations of reference-free XRF using monochromatized SR as the incident excitation radiation. In the first UHV instrumentation for reference-free XRF, the solid angle of detection is defined by a well-known diaphragm placed in front of the calibrated SDD detector at a given distance with respect to the center of the sample. In order to take advantage of the linear polarization of SR for reducing spectral background, the angle between the incident radiation and the fluorescence detection channel is 90°. When simultaneously detecting fluorescence radiation with two SDDs, normal incidence (θ = 90°) and observation angles ψ of the detectors of 30°, 45° or 60°, with respect to the sample surface, are chosen. θ_crit_ designates the critical angle of external total reflection at a flat sample surface. Calibrated photodiodes are employed to determine the incident radiant power or intensity of the excitation radiation.

**Figure 2 nanomaterials-12-02255-f002:**
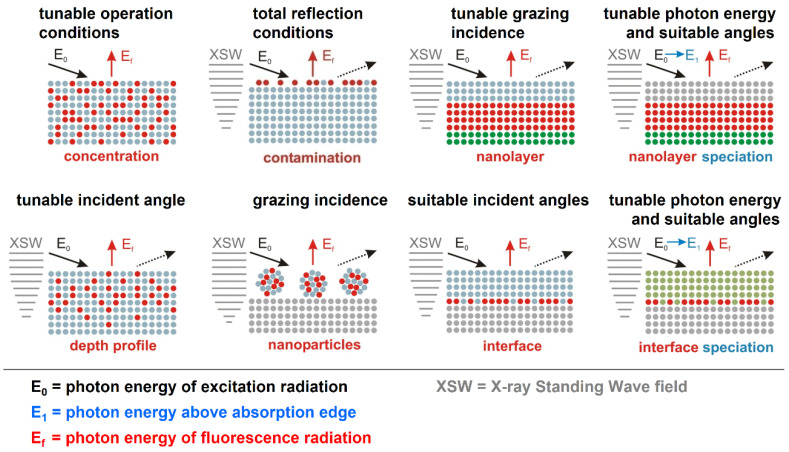
Modifications of SR-based XRS operational conditions for various classes of nanomaterials. The analytical discrimination capability, detection sensitivity, quantification reliability and information depth (driven by the penetration depths) can be favorably modified by an appropriate choice of operational parameters such as the photon energy of the incident radiation and a suitable set of incident angles.

**Figure 3 nanomaterials-12-02255-f003:**
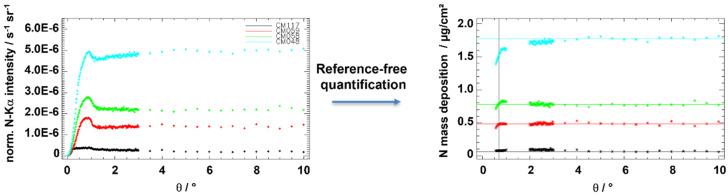
Experimentally revealed fluorescence intensities and deduced elemental mass depositions of different nitrogen-containing environmental samples.

**Figure 4 nanomaterials-12-02255-f004:**
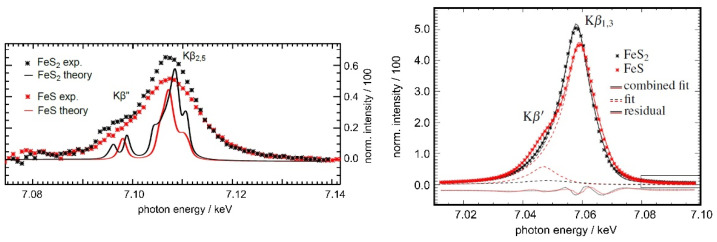
XES spectra of the iron sulfide compounds FeS and FeS_2_ recorded by the full-cylinder von Hamos spectrometer. The measured and fitted data of the Kβ and valence-to-core X-ray emission lines are shown to the left and right, respectively. A spectral comparison between experimentally determined Kβ″ and Kβ_2,5_ emission lines and related calculations using the OCEAN BSE code for core-level spectroscopy are depicted to the right.

**Figure 5 nanomaterials-12-02255-f005:**
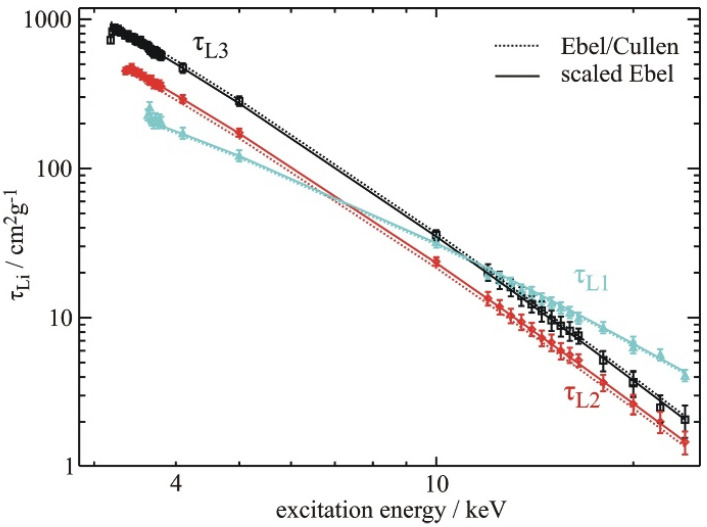
Experimentally determined PCS for the three Pd-L subshells as compared to theoretical Ebel/Cullen data. For the sake of comparison, the Ebel data have been scaled by 0.95 for the L3, 1.08 for the L2, and by 1.03 for the L1 PCS data, respectively.

**Figure 6 nanomaterials-12-02255-f006:**
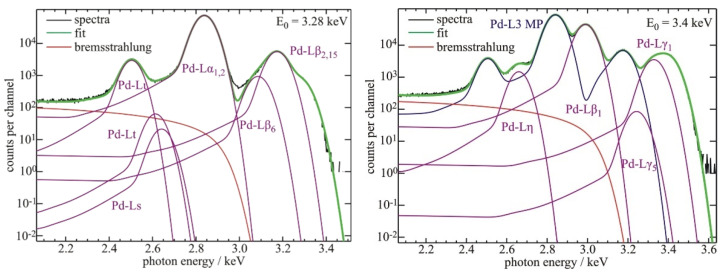
Response-function-based deconvolution of a 250 nm thick Pd layer XRF spectrum for each L-shell used for the experimental determination of Pd-L PCS. To the left, the excitation photon energy of 3.28 keV is chosen to be between the L3 and L2 sub-shell absorption edges and, to the right, the excitation photon energy of 3.4 keV lays between the L2 and L1 absorption edges. MP designates the L3 fluorescence line multiplet used as an ensemble in the L2-related spectral deconvolution.

**Figure 7 nanomaterials-12-02255-f007:**
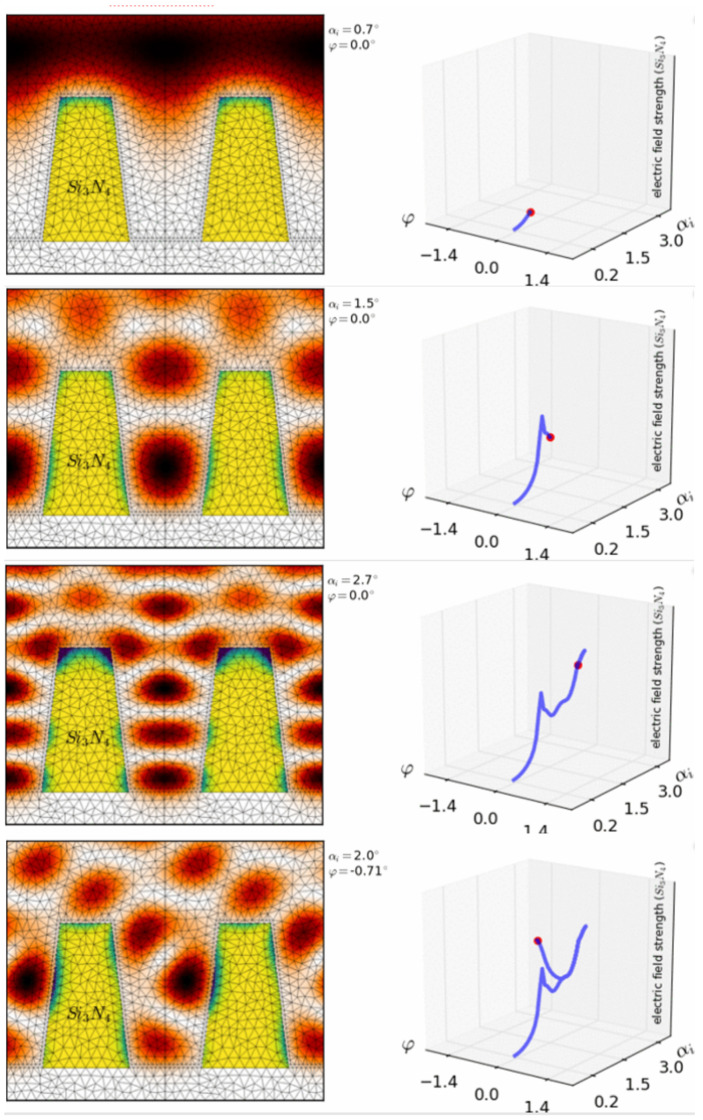
Color-coded intensity of the spatial distribution of the XSW values (left) within a regular nanostructure for four different GIXRF operational parameter combinations of the angle of incidence α_i_ and of the azimuthal angle φ between the nanostructure’s main orientation and the incident beam direction.

**Figure 8 nanomaterials-12-02255-f008:**
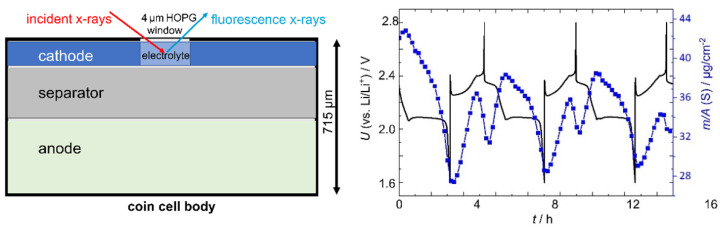
Set-up of a 715 µm thick coin cell battery probing S-K-related XRF and XAFS signals through a 4 µm thick HOPG entrance window under operation conditions (**left**). The elemental mass deposition *m*/*A* of sulfur (S) in dissolved PS is depicted in blue for the cathode side for the first three full cycles (**right**), while the electrochemical performance is depicted in black, corresponding to a percentage of lost cathodic sulfur ranging from about 15% to 25%.

## Data Availability

Not applicable.

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
