# Peer review of "Traceable Characterization of Nanomaterials by X-ray Spectrometry Using Calibrated Instrumentation"

_nanomaterials, 2022, doi:10.3390/nano12132255_

Round 1

Reviewer 1 Report

This manuscript reviews several x-ray spectroscopy related methods for traceable characterizations for nanoscopic materials, such as elemental concentration on nanometer scales, nanostructure (nanoparticles), etc. Overall, it was well written with many references. I think it is publishable given the following two issues are addressed.

  • In the section on X-ray fluorescence analysis, the paper focused on TXRF, GIXRF, and conventional XRF. However, another popular extension of XRF: scanning-free grazing-exit XRF (optical reciprocal to GIXRF) has been largely ignored. The author only mentioned briefly the concept later in another section. Fluorescence XSW effect also occurs for GEXRF for many situations in Figure 2. For completeness of the survey on XRF analysis, GEXRF deserves its own seat in section 2 with more references such as earlier work Tsuji et al., Spectrochimica Acta B. At. Spectrosc. 54, 1881 (1999), more recent work Baumann, et al., Anal. Chem. 89, 1965 (2017), as well as some recent advancements such as combining GIXRF and GEXRF simultaneously for high-resolution nanostructure measurement of thin films Jiang et al., Nature Comm. 11, 3197 (2020)
  • I am not sure if this manuscript only reviews techniques available at PTB. The author probably knows that there are several other STXM instruments with XRF capabilities around the synchrotrons worldwide. Many can achieve tens of nanometer resolutions, much higher than PTB. Examples are I08-SXM at Diamond, 29-ID-2 Soft X-ray Nanoprobe at NSLS-II, 26-ID-C Hard X-ray nanoprobe at APS, etc.

Author Response

Reply to reviewer #1:

The aim of this review is to demonstrate the absolute quantification capabilities of physically (SI-)traceable x-ray 
spectrometry based upon calibrated instrumentation and good knowledge on x-ray interaction probabilities (fundamental parameters) in 
different experimental configurations with respect to nanomaterials' characterisations. In this respect, the intention of this review is to 
complement other surveys on the various x-ray spectroscopy techniques focusing on discrimation capabilities such as detection 
sensitivities, spatial or energetic resolutions or the description of the related methodological principles without reiterating all of 
these other articles. 

The comments of the reviewer are very much appreciated. In view of his suggestions short introductory statements with extensive sets of 
relevant references have been added with respect to both the GEXRF and STXM-XRF works of several other groups as suggested by the reviewer.

Reviewer 2 Report

The article covers the capabilities of PTB in characterization of nanomaterials by x-ray spectrometry.  It goes in great detail explaining how the characterization is done if there is no reference materials by having well-calibrated and set up devices and beamline, as well as theoretical use of models, like monte carlo simulations.  It gives several examples of how these measurements are done and evaluated.

Overall, the paper presents no new groundbraking findings or methodologies, but is a good review of the capabilities of the PTB nanostructure characterization toolbox.  As a reference document for these efforts, it is satisfactory.

Author Response

Reply to reviewer2:

The aim of this review is to demonstrate the absolute quantification capabilities of physically (SI-)traceable x-ray 
spectrometry based upon calibrated instrumentation and good knowledge on x-ray interaction probabilities (fundamental parameters) in 
different experimental configurations with respect to nanomaterials' characterisations. In this respect, the intention of this review is to 
complement other surveys on the various x-ray spectroscopy techniques focusing on discrimation capabilities such as detection 
sensitivities, spatial or energetic resolutions or the description of the related methodological principles without reiterating all of 
these other articles. 

The comments of the reviewer are very much appreciated. Some scientific and technical statements were added to the manuscript 
as well as additional references on von Hamos spectrometer, GEXRF and STXM-XRF works of several other groups were included 
as requested by other reviewers.

Reviewer 3 Report

I send my comments in attachment

Author Response

Reply to reviewer 3:

The comments of the reviewer are very much appreciated and will be addressed in the following along the different categories raised:

a. The quality of all figures has been considerably enhanced. Some of the figures have been completely modified to underpin relevant aspects.

b. The number of references of other groups have been substantially increased as proposed also by another reviewer. Several references on 
relevant von Hamos spectrometer, GEXRF and STXM-XRF works were included. This changes the ratio of own publications underpining the review character 
of the publication as against the cited publications of other groups considerably as suggested. This procedure has been selected to meet the requirements of all reviewers.
The selected own publications are in line with the author's intention for this review: The aim of this review is to demonstrate the absolute quantification capabilities of physically (SI-)traceable x-ray 
spectrometry based upon calibrated instrumentation and good knowledge on x-ray interaction probabilities (fundamental parameters) in 
different experimental configurations with respect to nanomaterials' characterisations. In this respect, the intention of this review is to 
complement other surveys on the various x-ray spectroscopy techniques focusing on discrimation capabilities such as detection 
sensitivities, spatial or energetic resolutions or the description of the related methodological principles without reiterating all of 
these other articles.

c. The minor comments of the reviewer have been all addressed. In particular, the description of the Sherman equation quantities has been 
improved as suggested. 

d. Additional technical details about the von Hamos spectrometer have been added at several paragraphs as suggested by the reviewer.

Round 2

Reviewer 3 Report

The authors replied to all the comments